# Proteomic characterization of GSK3β knockout shows altered cell adhesion and metabolic pathway utilisation in colorectal cancer cells

Emily Bowler-Barnett[¤a], Francisco D. Martinez-Garcia[¤b], Matthew Sherwood,
Ahood Aleidan, Steve John, Sara Weston, Yihua Wang, Nullin Divecha, Paul Skipp,
Rob M. Ewing *

School of Biological Sciences, Faculty of Environmental and Life Sciences, University of Southampton, Southampton, United Kingdom

¤a Current address: European Bioinformatics Institute, EMBL-EBI, Cambridge, United Kingdom
¤b Current address: Department of Pathology and Medical Biology, University Medical Center Groningen, University of Groningen, Groningen, the Netherlands
* rob.ewing@soton.ac.uk

**Data Availability Statement:** The mass spectrometry proteomics data have been deposited to the ProteomeXchange Consortium via the

## Abstract

Glycogen-specific kinase (GSK3β) is an integral regulator of the Wnt signalling pathway as well as many other diverse signalling pathways and processes. Dys-regulation of GSK3β is implicated in many different pathologies, including neurodegenerative disorders as well as many different tumour types. In the context of tumour development, GSK3β has been shown to play both oncogenic and tumour suppressor roles, depending upon tissue, signalling environment or disease progression. Although multiple substrates of the GSK3β kinase have been identified, the wider protein networks within which GSK3β participates are not well known, and the consequences of these interactions not well understood. In this study, LC-MS/MS expression analysis was performed using knockout GSK3β colorectal cancer cells and isogenic controls in colorectal cancer cell lines carrying dominant stabilizing mutations of β-catenin. Consistent with the role of GSK3β, we found that β-catenin levels and canonical Wnt activity are unaffected by knockout of GSK3β and therefore used this knockout cell model to identify other processes in which GSK3β is implicated. Quantitative proteomic analysis revealed perturbation of proteins involved in cell-cell adhesion, and we characterized the phenotype and altered proteomic profiles associated with this. We also characterized the perturbation of metabolic pathways resulting from GSK3β knockout and identified defects in glycogen metabolism. In summary, using a precision colorectal cancer cell-line knockout model with constitutively activated β-catenin we identified several of the diverse pathways and processes associated with GSK3β function.

PRIDE partner repository with the dataset identifier PXD014632.

**Funding:** RME acknowledges Medical Research Council (MR/S01411X/1) and European Commission FP7 ("Oncoprotnet") for funding; YW acknowledges Medical Research Council (MR/S025480/1). FDMG and RME acknowledge support from the Mexican National Council for Science and Technology (CONACyT)and Technology (CONACyT) The funders had no role in study design, data collection and analysis, decision to publish, or preparation of the manuscript.

**Competing interests:** The authors have declared that no competing interests exist.

## Introduction

Glycogen synthase kinase-3 beta (GSK3β) is a serine/threonine kinase in the glycogen synthase kinase subfamily with diverse roles and interaction partners in a wide range of signalling pathways and cellular functions [1]. Wnt/ β-catenin signalling is a principle driver pathway in colorectal cancer, with evidence of mutations and aberrant signalling from an early stage of tumour development. Following pre-phosphorylation by casein kinase 1 (CK1), GSK3β phosphorylates β-catenin at serine residues 33, 37, and 41 which targets β-catenin for proteosomal degradation [2]. When Wnt signalling is activated by the family of Wnt ligands binding to receptors, such as Frizzled, GSK3β and CK1 phosphorylate low-density lipoprotein receptor-related protein 6 (LRP6) at conserved PPSPXS motifs, which then initiates the recruitment of the scaffold protein Axin. It is this interaction with the Axin/LRP6 complex that surrounds GSK3β with phosphorylated residues and directly inhibits its ability to phosphorylate β-catenin [3, 4]. Following inhibition of GSK3β, β-catenin accumulates in the cytosol and then undergoes nuclear import where it activates target transcription factors and gene transcription [4].

Although GSK3β plays a key role in regulating Wnt/ β-catenin signalling activity, its role in other oncogenic or tumour suppressor pathways is less well understood. Whether GSK3β plays a primarily oncogenic or tumour suppressor role appears to depend on the cellular context and tumour type. GSK3β has been shown to be expressed at lower levels in skin cancer tumours and knockdown of GSK3β in breast cancer mammary models results in activation of Wnt signalling and the formation of adenosquamous carcinomas [5, 6]. GSK3β has been shown to be significantly increased in non-small cell lung cancer patient samples with increased expression correlated with a poor long-term prognosis, and GSK3β was also shown to have inconsistent effects on Wnt/ β-catenin signalling depending on cell-type [7].

GSK3β has been linked to several roles in the regulation of cell adhesion and migration in keratinocytes. For example, desmoplakin plays a key role in the formation of intercellular junctions required for skin and heart integrity, and desmoplakin is post-translationally modified by GSK3β [8]. GSK3β is also known to phosphorylate Focal Adhesion Kinase resulting in a reduction in its activity and in cell motility [9]. Many studies have linked GSK3β to the proliferation of cardiomyocytes and to the fibrotic remodelling that occurs in the ischemic heart, further implicating GSK3β in the remodelling of cell-cell junctions [10].

In this study, we sought to profile the proteomic and phenotypic effects of GSK3β in Wnt/ β-catenin-dependent cancer cells to identify other GSK3β-dependent pathways and processes in addition to canonical Wnt signalling. We used an isogenic pair of colorectal cancer cell lines (HCT116-GSK3β-WT and HCT116-GSK3β-KO) and showed that canonical Wnt signalling is largely unimpaired by complete knockout of GSK3β whereas specific and dramatic effects on cell-cell adhesion proteins and metabolic pathways were observed.

## Methods

### Maintenance of cell lines

HCT116-GSK3β-KO and paired isogenic control cell line HCT116-GSK3β-WT are commercially (Cellectis) created cell-lines (created by integrating a ssDNA oligonucleotide containing several stop codons at the GSK3 β locus with TALEN technology). These cell lines were cultured in McCoys 5A culture medium (Gibco) supplemented with 10% FBS (Gibco) and 1% PenStrep (Gibco) and maintained in 5% $CO_2$ at 37˚C with twice weekly passaging. For passaging, cells were disassociated from the flask following warm DPBS (Gibco) wash using 0.5ml of warm TrypLE Express enzyme 1X (ThermoFisher Scientific), and incubated for five to ten

**Table 1. List of antibodies used in this study.**

| Antibody | Product code | Vendor | Host species | Dilution |
|---|---|---|---|---|
| Actin | 6H10D10 | Cell signaling | Rabbit | 1:1000 |
| Axin2 | D48G4 | Cell signaling | Rabbit | 1:1000 |
| Beta-Catenin | MA1-301 | ThermoFisher | Mouse | 1:1000 |
| Dvl2 | 3126S | Cell signaling | Rabbit | 1:1000 |
| Dvl3 | 3218S | Cell signaling | Rabbit | 1:1000 |
| Gsk3Beta | 27C10 | Cell signaling | Rabbit | 1:1000 |
| Jup | 75550S | Cell signaling | Rabbit | 1:1000 |
| Pkp2 | PA5-53144 | ThermoFisher | Rabbit | 1:1000 |
| Pkp3 | 35–7600 | ThermoFisher | Mouse | 1:1000 |
| Tcf3 | 2883S | Cell signaling | Rabbit | 1:1000 |
| Tcf4 | 2569S | Cell signaling | Rabbit | 1:1000 |

minutes. Disassociated cells were centrifuged at 1,000 x g for five minutes before reconstitution in warm McCoys 5A (Gibco) medium before seeding a tenth of the culture population in a fresh culture flask. Whole cell protein lysate extract was performed by resuspension of cells in 0.1M TEAB, 0.1% SDS and lysed by sonication on ice. The suspension was then centrifuged at 13,000 x g, for 20 minutes at 4˚C, supernatant was then removed and protein concentration measured by a DirectDetect® spectrometer.

## SDS-PAGE and immunoblotting

Equal amounts (15 μg) of proteins from different samples were loaded on 8% acrylamide gel and subjected to electrophoresis (120 volts, 90 minutes). Afterwards, proteins were transferred to nitrocellulose membrane (80 volts, 0.4 Amps, 3 hours) (Whatman 10402594, Dassel, Germany) and blocked for 30 minutes using a 5% milk solution. Membranes were then incubated at 4˚C and proteins identified using near-infrared fluorescent secondary antibodies (Li-Cor) and imaged using a Li-Cor Odyssey® CLx with Image Studio Lite V5.2 software. Table 1 lists all antibodies used.

## TopFlash assay measurement of Wnt signalling activity

Cells were seeded in 24 well plates at a density of $1 \times 10^5$ cells per well and cultured for 24 hours in McCoys 5A culture medium (Gibco) supplemented with 10% FBS (Gibco) and 1% PenStrep (Gibco) in 5% CO2 at 37˚C. Transfection was then carried out by Lipofectamin® 3000 (ThermoFisher Scientific) according to manufacturers' protocols. We performed Top-FLASH assays as previously described [11]. TopFLASH assay constructs used; M50 Super 8x ToPFLASH (AddGene), M51 Super 8x FopFLASH (AddGene), and pcDNA3.1-ccdB-Renilla (AddGene). Cells were cultured in transfection media for 6 hours before transfection media was removed and fresh growth media was added to cells. Cells were then cultured overnight before measurement of luciferase and renilla reporter signals using a Dual-Luciferase® Reporter assay system (Promega) and a GloMax® Discover Microplate reader (Promega) according to manufacturers' protocols. Results were then normalized to renilla internal transfection control, and then plotted graphically using GraphPad Prism V7.

## Metabolic profiling of HCT116-GSK3β-WT and HCT116-GSK3β-KO cells

Metabolism profiling was carried out by Agilent Seahorse XFp Cell Energy Phenotype test kit on a Seahorse XFp analyser (Agilent.com). Protocols were carried out according to

manufacturer guidelines and data was analysed using Agilent Seahorse Wave desktop V2.5 https://www.agilent.com/en/solutions/cell-analysis). Metabolic pathway analysis was performed using Escher [12] in conjunction with the BiGG database [13].

## Mechanical stress test of cell-cell junctions

Cell-cell adhesion strength was assessed using dispase cell-substrate detachment enzyme on cell monolayers followed by mechanical stress test, integrity of cell monolayer was then assessed via light microscopy.

HCT116-GSK3β-WT and HCT116-GSK3β-KO cells were seeded in 6-well culture plates (Corning) and grown to confluence. Media was removed from cultures and cell monolayer washed with Hanks balanced salt solution (HBSS) (14025092 ThermoFisher Scientific). Cell monolayers were then incubated with dispase enzyme (2.5 Units/ml in HBSS) (D4693 Sigma) at 37˚C for 30 minutes or until monolayers detached from plate bottom. Detached monolayers were then imaged before mechanical stress test with a Leica M2 16 F microscope and S Viewer V3.0.0 software. Mechanical stress was then induced used an orbital shaker (Labscale) for 3 minutes at 300rpm (orbit of 16mm). Cell monolayer following mechanical stress was then assessed for intactness.

## Cellular migration assay

HCT116-GSK3β-WT and HCT116-GSK3β-KO cells were seeded in 6-well culture plates (Corning) and grown to confluence. Cells were then treated with 2mM thymidine (Sigma) supplemented media for 24 hours prior to initiation of scratch assay. Following scratch of cell monolayer, media was aspirated and cell debris was removed by washing twice with 2mM thymidine supplemented media. Culture of cells was then continued in 2mM thymidine supplemented media with migration into scratch assessed at time points over a period of 22 hours via imaging using an EVOS XL core cell imaging system (ThermoFisher Scientific), and migration into scratch area measured by ImageJ software.

## Growth and colony formation analysis

$1 \times 10^5$ HCT116-GSK3β-WT and HCT116-GSK3β-KO cells were seeded in 6 well plates (Corning) on day 0, and maintained in McCoys 5A culture medium (Gibco) supplemented with 10% FBS (Gibco) and 1% PenStrep (Gibco), 5% $CO_2$ at 37˚C. Cells were disassociated using warm TrypLE Express enzyme 1X (ThermoFisher Scientific) and counted using a haemocytometer counting chamber (Neubauer improved) every 24 hours. For colony growth analysis, $5 \times 10^3$ cells were seeded per well (12 well plate (Corning)) in a total of 500µl of 0.35% agar supplemented complete McCoys 5A medium (10% FBS, 1% PenStrep), the cultures were incubated at room temperature for 3 minutes and then incubated for 5 minutes at 4˚C before continued culture at 37˚C, 5% $CO_2$ (ThermoFisher Scientific) as described previously. Cell colonies were counted manually following visualization of the cultures using a light microscope (EVOS XL, ThermoFisher Scientific). Cell counts were recorded every 24 hours for 6 days following seeding.

## Proteomic profiling and mass-spectrometry

Cell pellets ($3 \times 10^6$) were lysed (0.1M TEAB, 0.1% SDS) with pulsed sonication. Methanol/chloroform extraction was performed on 100ug of protein for each lysate and pellet finally redissolved in 100ul of 6M Urea (Sigma), 2M thiourea (Sigma), 10mM HEPES buffer (Sigma), pH7.5. Samples were reduced (1M DTT), alkylated (5.5M iodoacetamide) then diluted in

400µl of 20mM ammonium biocarbonate (Sigma) and digested with trypsin (Promega) (1/50 w/w) overnight. As an internal quantification standard, 150 fmol of enolase (*Saccharomyces cerevisiae*) (Waters) and Hi3 *Escherichia coli* standard (Waters) were added to each sample. Samples were 12xfractionated (Agilent 3100 OFFGEL fractionator) in IPG buffer, pH 3–10 (GE Healthcare) diluted 1:50 in 5% glycerol, in 13cm IPG strips, pH 3–10 (GE Healthcare) and peptides focused for 20 kVh. Each fraction was acidified to <3.0 using trifluoroacetic acid (TFA) (Sigma) and solid phase extraction performed using Empore C18 96-well solid phase extraction plate (Sigma). Samples were eluted in 150µl of 80% acetonitrile, 0.5% acetic acid, lyophilised and then stored at -20˚C until re-suspension in 10µl 98% dH$_2$O/acetonitrile and 0.1% formic acid prior to use. Nano-LC separation was performed using the nanoAcquity UPLC 2G Trap column system (Waters) at a rate of 5ul per min and washed with buffer A (98% dH$_2$O/acetonitrile + 0.1% formic acid) for 5 minutes. Peptides were separated on an Acquity UPLC Peptide BEH C18 column (130Å, 1.7µm, 2.1mm x 150mm, 1/pkg (Waters)) over a 90-minute linear gradient of 5% to 40% with a flow rate of 0.3µl/min with buffer B (80% acetonitrile/dH$_2$O + 0.1% formic acid) and completed with a 5 minutes rinse with 85% buffer B, at a flow rate of 300nl/min. Mass Spectrometry analysis was performed on a Synapt G2-S HDMS system (Waters). Samples were sprayed directly using positive mode-ESI, and data was collected in MS$^E$ acquisition mode, alternating between low (5v) and high (20-40V) energy scans. Glu-fibrinopeptide (*m/z* = 785.8426, 100 fmol/µl) was used as LockMass, and was infused at 300 nl/min and sampled every 13 seconds for calibration.

## Mass-spectrometry data processing and analysis

Raw data files were processed using ProteinLynx Global Server (PLGS, Waters) version 3.0.2, and optimal processing parameters were selected by Threshold Inspector software (Waters). Data-base search (Uniprot Human reference proteome; UP000005640—July 2017 + yeast enolase P00924) was carried out using the Ion-accounting algorithm in PLGS (Waters), parameters included a false discovery rate of 4%, a maximum protein size of 500,000 Da, and trypsin cleavage with an allowable error of 1 missed cleavage with oxidation of methionine, and carbamidomethylation of cysteine modifications. For protein quantification, the Hi3 calculation method of using the top 3 most intense tryptic peptides of both the internal standards and of each protein was used. Only quantified proteins were included in further analysis, therefore all had at least 3 peptides present, indicating confidence in the identification of the protein from the spectra. Protein quantification was normalised by the sum of the quantification of total proteins per sample, and multiplied by $1x10^6$, the result of which was $log_2$ and averaged between replicated samples. The average for each protein within a sample was then used to calculate fold change in the expression of the proteins identified in both the HCT116-GSK3 β-WT and HCT116-GSK3 β-KO cells. A threshold value of a 2-fold change in protein expression was applied. Gene ontology (GO) analysis was performed using Gorilla (Gene Ontology enRIchment anaLysis and visuaLizAtion tool) in conjunction with GO terms, to visualize significant enrichment of the datasets against the entire human proteome, results were then visualized in REVIGO, in the TreeMap format [14, 15]. GO analysis was also performed in PANTHER to summarise the enrichment of GO terms in biological pathways of the whole dataset against the human proteome [16] and using the Pathway Studio software (Elsevier) v9.0. Protein interaction analysis was performed using STRING v10.0, a database collating protein interaction information, including experimental evidence and context and co-expression, to infer a network of protein-protein interaction partners within the dataset. The mass spectrometry proteomics data have been deposited to the ProteomeXchange Consortium via the PRIDE partner repository with the dataset identifier PXD014632.

## Results

### Proteomic analysis of knockout GSK3β colorectal cancer cells

As a model for GSK3β knockout, we selected Talen-induced knockout cells (HCT116-GSK3β-KO) and a paired isogenic control cell line (HCT116-GSK3β-WT) in the colorectal cancer cell-line HCT116. To understand the role of GSK3β knockout in a Wnt activated (β-catenin mutant) environment we carried out whole cell proteomic profiling of HCT116-GSK3β-WT and HCT116-GSK3β-KO cells (Fig 1A). HCT116 cells carry the oncogenic stabilized allele (ΔS45) of β-catenin that lacks the Serine residue at position 45 that is normally phosphorylated by CK1. This priming normally allows GSK3β to further phosphorylate β-catenin leading to proteasomal degradation. LC-MS/MS analysis of the wild type and knockout cells identified 6,282 proteins, 1200 of which were quantified and identified across all samples (S1 File). We identified a set of 440 proteins that were uniquely detected in HCT116-GSK3β-KO or HCT116-GSK3β-WT cells or significantly ($p < 0.05$; Student's T-test) differentially expressed in one of the cell lines. Gene Ontology (GO) enrichment analysis of the differentially expressed set of proteins revealed significantly enriched GO terms including "metabolic process' (rank = 6; p-value = 3e-17). Enrichment analysis of pathways identified 'Tricarboxylic Acid Cycle' and 'Glycogen Degradation' (ranked 3rd and 4th) as highly significant. We also observed that cell-cell adhesion associated terms and pathways were significantly enriched, including 'Actin-based cytoskeleton assembly (ranked 6th), 'focal adhesion assembly' (ranked 10th) and 'hemidesmosome assembly' (ranked 15th), we, therefore, focused on further analysis of these functional roles (see S2 File for pathway analysis).

### Wnt/ β-catenin signalling is not perturbed in GSK3 β knockout cells

Although the role of GSK3β in regulation of canonical Wnt signalling is well known, we first sought to characterize the HCT116-GSK3β-KO cells by analysis of key Wnt signalling proteins as shown in Fig 2. Although β-catenin protein levels are directly regulated through serine phosphorylation by GSK3β and subsequent proteasomal degradation, no significant differences were observed in β-catenin protein expression between HCT116-GSK3β-KO and HCT116-GSK3β-WT cells (Fig 2C). This finding is consistent with the understanding that GSK3β cannot regulate oncogenic β-catenin through phosphorylation because of the loss of the CK1 phosphorylation site at Serine 45. We also analysed the β-catenin destruction complex component Axin (AXIN2), the Wnt regulatory protein Dishevelled (DVL2 and DVL3), and the transcription factors Transcription Factor 3 (TCF3) and Transcription factor 4 (TCF4) that drive the Wnt transcriptional response (Fig 2D–2F), Western blot analysis shows there are no significant changes to the abundance of these key Wnt components in the absence of GSK3β. We tested the canonical Wnt-driven transcriptional activity of these cells through reporter (TopFLASH) analysis as shown in Fig 2G. TopFLASH reporter assays showed that both HCT116-GSK3β-KO and HCT116-GSK3β-WT had robust levels of TCF/ β-catenin-driven reporter activity and no statistically significant differences in the levels of that activity. Finally, we identified additional Wnt signalling components in the mass-spectrometry data (S1 File). β-catenin showed no significant difference between WT and KO cells, whilst APC (Adenomatous Polyposis Coli), an important component of the destruction complex was only identified in WT cells, indicating that loss of GSK3β may result in loss or reduced expression of APC. Finally, the Casein Kinase 1 alpha subunit (CSNK1A1) which phosphorylates β-catenin at S45 showed reduced expression in knockout cells (p-value = 0.013).

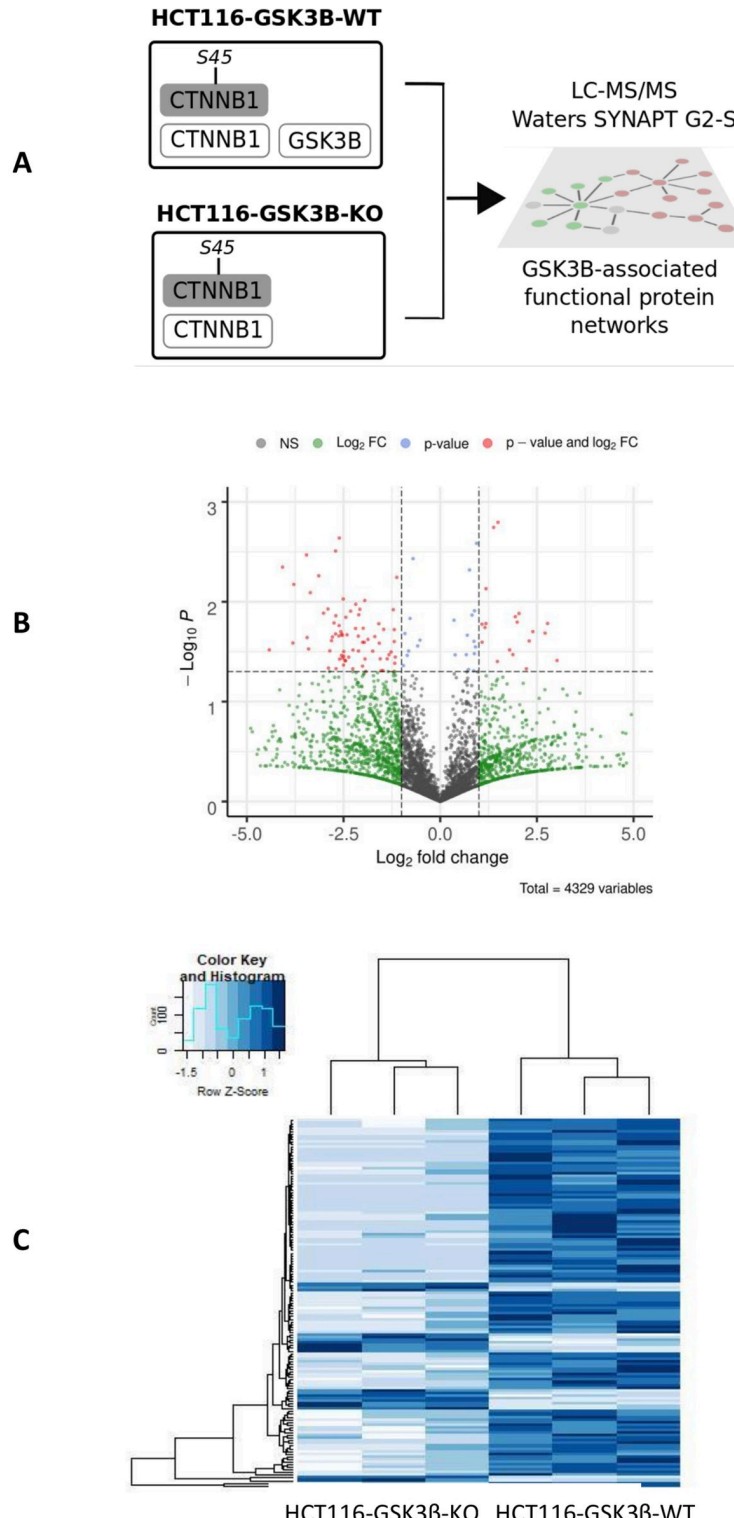

**Fig 1.** A. Overview of experimental design. HCT116 cells (with stabilizing β-catenin mutation) with wild-type (HCT116-GSK3β-WT) or knockoutGSK3β (HCT116-GSK3β-KO) were profiled using LC-MS/MS to identify associated protein networks. B. Volcano plot of protein expression from mass-spectrometry study. Plot indicates gene symbols for identified proteins with log2 fold-change greater than |2| and p-value < 0.05. C. Heat map of protein expression. Showing replicated wild-type and knockout samples showing those proteins with altered expression (with log2 fold-change greater than |2| and p-value < 0.05).

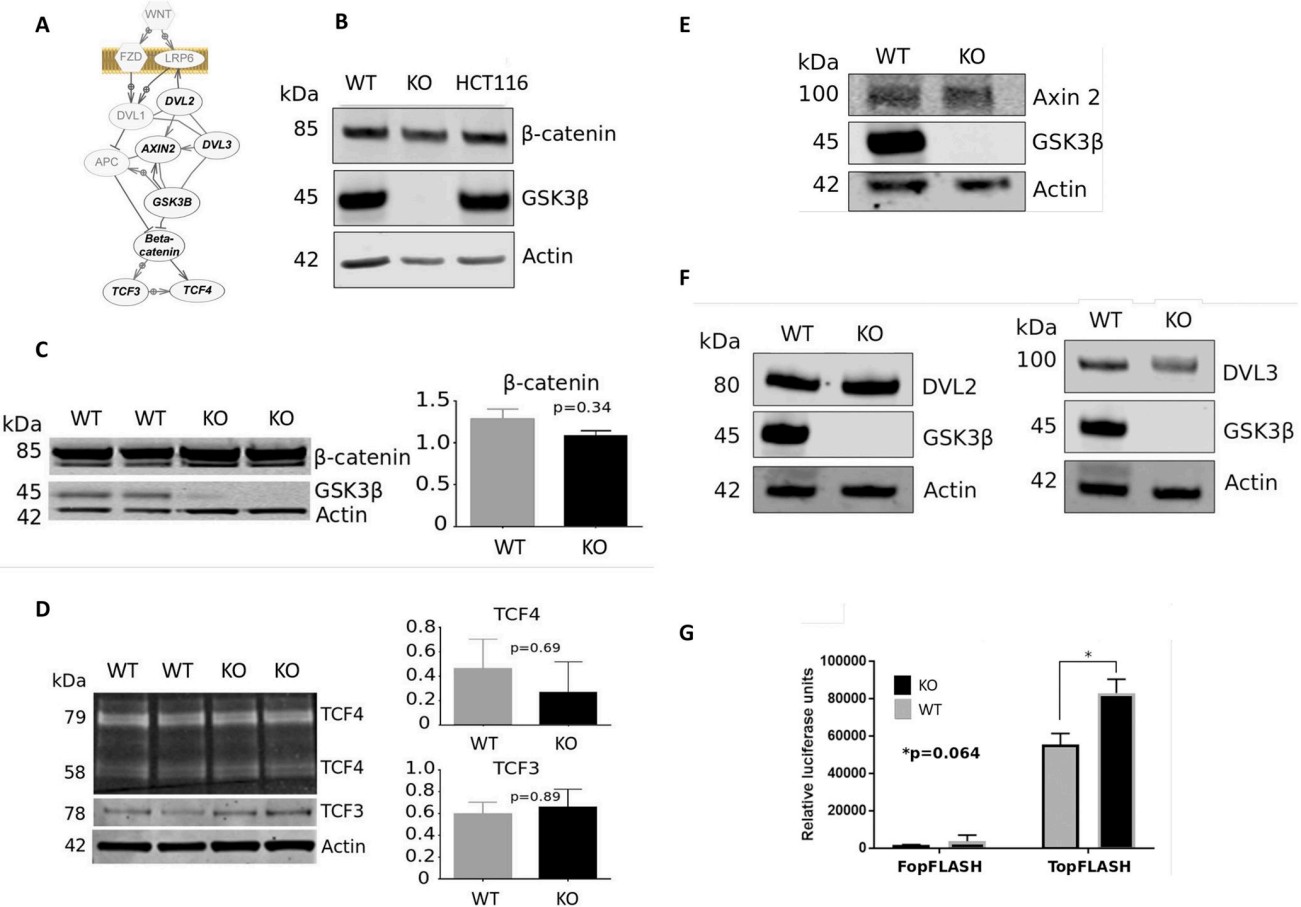

**Fig 2.** (**A**) Schematic of the canonical Wnt signalling pathway and summary of key proteins analysed in this study. Proteins identified in the mass-spectrometry study are shown, those not analysed by Western blot are shown in grey. Western blot analysis of canonical Wnt components and GSK3β protein expression characterization in cell models. 15 μg of total protein lysates from each cell-line were analyzed on a single phase 8% SDS gel and alpha-actin used as loading control. All were replicated in at least n = 3 unless indicated otherwise. Quantifications (arbitrary intensity units) are mean values ± SEM (Mann-Whitney U test p-value indicated). (**B**) Protein expression of GSK3β and β-catenin in HCT116 cells (independently sourced isolate), HCT116-GSK3β-KO cells and HCT116-GSK3β-WT cells. (**C**) β-catenin, (**D**) TCF3 and TCF4, (**E**) Axin 2, (**F**) DVL2 and DVL3 (n = 2). All Westerns are representative examples, and quantifications are expressed as arbitrary intensity units. (**G**) TopFLASH reporter assay of canonical Wnt signalling activity in HCT116-GSK3β-WT and HCT116-GSK3β-KO cells. Cells were transfected with TopFLASH plasmids, TopFLASH luciferase activity was assessed 48 hours post-transfection following passive lysis of cells. Luciferase signaling was normalized to Renilla transfection control. p-value 0.064, n = 3.

## Loss of GSK3β results in dysregulation of desmosome and hemidesmosome proteins

We analysed the enriched pathways in the differentially abundant proteins (Fig 3A) and found that pathways associated with cell-cell communication and adhesion were significantly enriched in the differentially abundant proteins identified. These include focal adhesion assembly (p = 0.005) and hemidesmosome assembly (p = 0.009). The HCT116-GSK3β-KO cells showed a specific reduction in protein abundance of desmoplakin (DSP), plakoglobin (JUP) and desmoglein (DSG), and an increase in abundance of their essential linker proteins, plakophilins (PKP2, PKP3), as well as altered expression of integrin family members integrin subunit alpha 6 (ITGA6) and integrin subunit beta 4 (ITGB4). ITGA6 and ITGB4 which mediate attachment between cells and of cells to the extracellular matrix. Western blot analysis was used to analyse the abundance of the plakophilins, plakoglobin and desmoglein and we found

**A**

| Pathway | p-value |
|---|---|
| Translation | 1.18E-12 |
| Chromatin remodeling | 3.95E-06 |
| mRNA processing | 4.44E-06 |
| Actin-based cytoskeleton assembly | 3.39E-05 |
| Tricarboxylic acid cycle | 8.91E-05 |
| Nucleolus organization and biogenesis | 3.15E-04 |
| Purine metabolism | 3.75E-04 |
| Spindle assembly | 1.16E-03 |
| Polymerase I transcription | 4.58E-03 |
| Focal junction assembly | 4.79E-03 |
| Presentation of endogenous peptide antigen | 5.22E-03 |
| Glycogen degradation | 6.23E-03 |
| Chromosome condensation | 7.21E-03 |
| Microtubule cytoskeleton assembly | 7.64E-03 |
| Hemidesmosome assembly | 9.49E-03 |

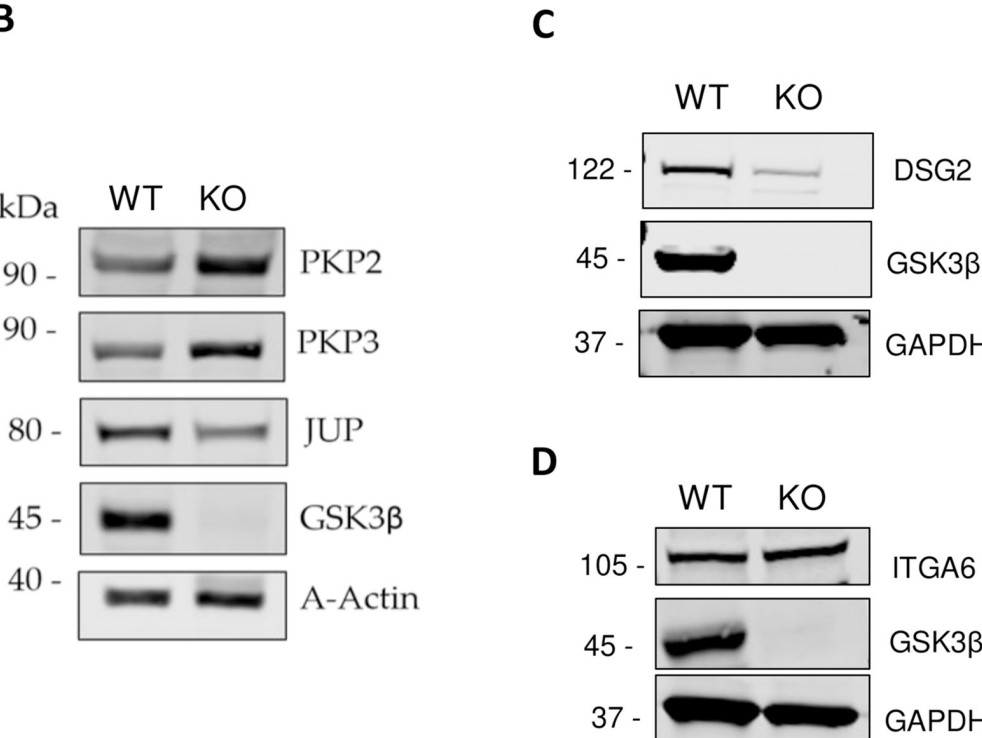

**Fig 3.** A. Enriched pathways represented in differentially abundant proteins. Enriched pathways were identified using Pathway Studio from the set of differential proteins (p<0.05). Pathways with significance p < 0.01 are shown and include pathways involved in focal adhesion, hemidesmosomes as well as glycogen metabolism. B. Western blot analysis and mass spectrometry analysis of cell-cell adhesion proteins. 15 μg of total protein lysates from HCT116-GSK3β-WT and HCT116-GSK3β-KO cells were analysed by western blot analysis for protein expression of

GSK3β, PKP2, PKP3, and JUP. Protein abundance of Actin was used as a loading control. n = 3. C. Western blot
analysis and mass spectrometry analysis of cell-cell adhesion proteins. 15 μg of total protein lysates from
HCT116-GSK3β-WT and HCT116-GSK3β-KO cells were analysed by western blot analysis for protein expression of
DSG2 and GSK3β. Protein abundance of GAPDH was used as a loading control. n = 2. D. Western blot analysis and
mass spectrometry analysis of cell-cell adhesion proteins. 15 μg of total protein lysates from HCT116-GSK3β-WT and
HCT116-GSK3β-KO cells were analysed by western blot analysis for protein expression of ITGA6 and GSK3β. Protein
abundance of GAPDH was used as a loading control. n = 2.

distinct alterations to the abundance of these proteins between the HCT116-GSK3β-KO and
WT cells as shown in Fig 3B and 3C. We also analysed the abundance of ITGA6, identified in
the mass-spectrometry data as differentially abundant (p = 0.042), although the Western blot
did not reveal a significant difference between WT and KO cells (Fig 3D).

Given the role of desmosomes and hemidesmosomes in cell-cell adhesion and tissue struc-
ture, we investigated whether HCT116-GSK3 β-KO exhibited altered cell adhesion using a
mechanical stress assay as shown in Fig 4A. Following mechanical agitation of detached mono-
layers of HCT116-GSK3β-KO cells the cell monolayer fragmented into numerous pieces, in
contrast to the HCT116-GSK3β-WT cells in which the monolayer remained intact. This indi-
cates that loss of GSK3β and the dramatic perturbation of desmosome and hemidesmosome
proteins is associated with diminished cell-cell adherence and a reduction in mechanical
strength of these cell-cell junctions. Previous findings showed that loss of Plakophilins and
subsequent disruption of desmosomal plaques resulted in an increased migratory ability of
cells [17, 18]. We investigated the migration of HCT116-GSK3β-KO and HCT116-GSK3β-
WT cells over 24 hours on cell cycle arrested cells, as shown in Fig 4B, and found that the
knockout cells exhibited an increase in migration potential. These results mirror other findings
in which disruption of the desmosome complex, incurs a selective advantage on cells for
migration following disruption of the primary culture and which translates into an increased
metastatic ability in cancer models [19]. To investigate if this also affected the cells ability to
form colonies from single cells, a soft agar colony growth assay was performed and colony
number over 10 days recorded (Fig 4C). Results indicate that loss of GSK3β did not incur any
colony-forming advantage to HCT116-GSK3β-KO cells.

## Dysregulated carbon metabolism in GSK3β knockout cells

To investigate the significant enrichment of proteins involved in metabolic pathways, we
mapped the set of 440 differentially expressed proteins to metabolic pathways using the Escher
interface [12] as shown in Fig 5A. This analysis showed a marked distinction between the
modes of energy metabolism between the HCT116-GSK3β-KO and HCT116-GSK3β-WT cells
whereby the knockout cells had reduced levels of many of the proteins involved in aerobic car-
bon metabolism but an apparent elevation of ketogenesis. BDH1, the primary 3-Hydroxybuty-
rate Dehydrogenase, was substantially elevated in the knockout cells. BDH1 catalyses the
interconversion of acetoacetate and hydroxybutyrate, which are the two major ketone bodies
produced from fatty acids. As shown in Fig 5B, glycogen metabolism proteins were highly dys-
regulated between knockout and wild-type cells. The three human glycogen phosphorylase iso-
zymes (PYGM, PYGL and PYGB) function as semi-tissue-specific enzymes catalysing the
production of glucose from glycogen. All were significantly altered between knockout and
wild-type cells (PYGM p-value = 0.04; PYGL p-value = 0.003; PYGB p-value = 0.02). In addi-
tion, the Glycogen synthase 1 enzyme was elevated in wild-type cells (p-value = 0.001). To fur-
ther investigate these observations and their effects on cell growth and metabolism, we
performed a cell energy phenotype assay (Agilent Seahorse XFp). This indicated that
HCT116-GSK3β-KO cells have a reduced oxygen consumption rate (OCR) and extracellular

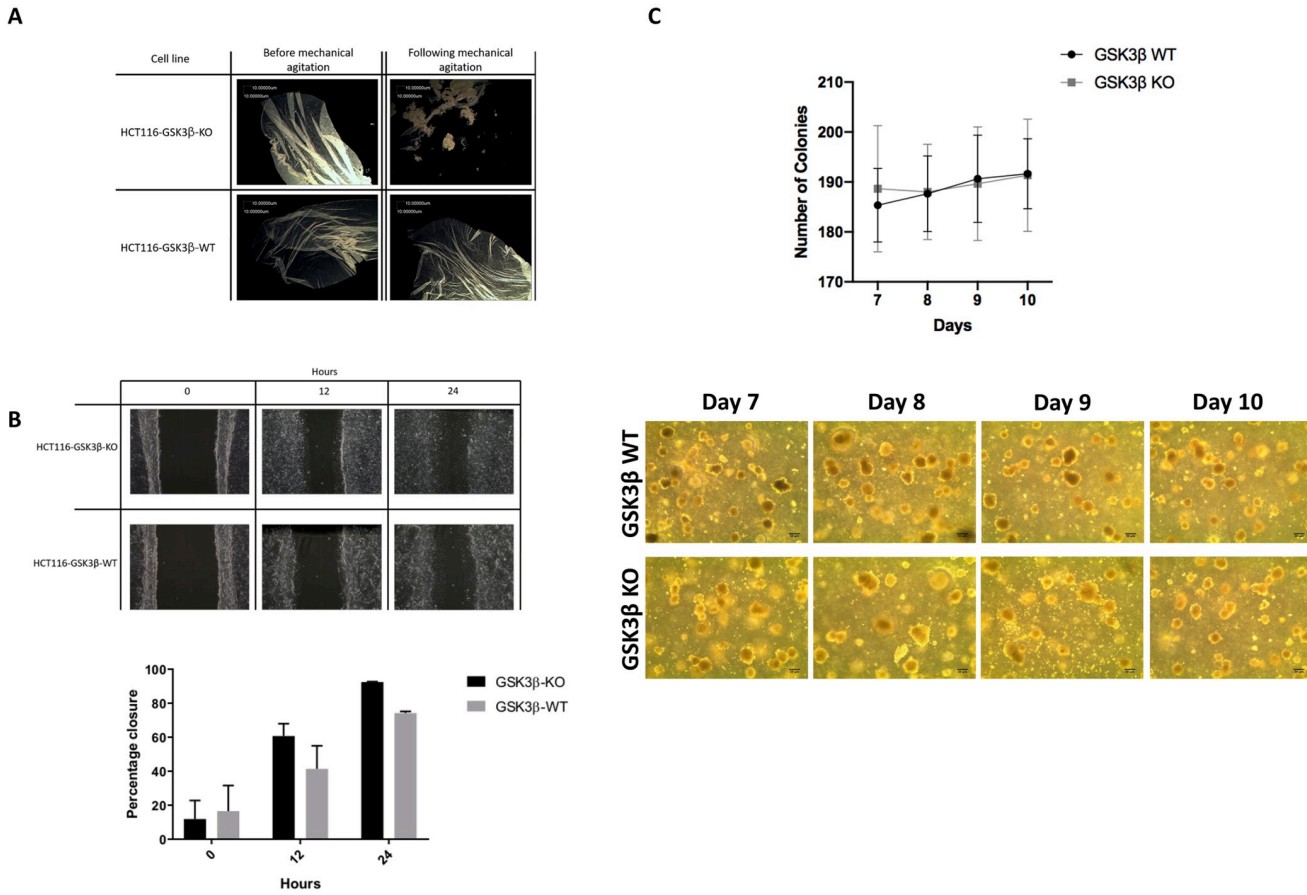

**Fig 4.** A. Mechanical stress test of cell-cell adhesion. Monolayer cultures of HCT116-GSK3β-WT and HCT116-GSK3β-KO cells were detached from culture surface and cell adhesions stressed with mechanical agitation. B. Cell migration assay of WT and KO cells. HCT116-GSK3β-WT and HCT116-GSK3β-KO cells were grown until confluent and then cell cycle arrested with 24-hour treatment of 2mM thymidine. Migration scratch was then made in the monolayer and capacity of cells to migrate into the free space assessed by light microscopy. Movement of cell migration front was measured as area free of cells and percentage migration calculated and presented graphically. Graphical representation of cell migration is presented as percentage closure of scratch. C. Soft agar colony growth assay Cells were seeded individually in soft agar matrix supplemented with complete growth media and colony formation was assessed every 24 hours over a 10-day period by light microscopy.

acidification rate (ECAR) when compared to HCT116-GSK3β-WT cells at both the basal level and under stressed conditions (Fig 5C). OCR was used as an indicator of mitochondrial respiration and ECAR was used as an indicator of glycolysis, the reduced rates of which when taken together indicate HCT116-GSK3β-KO cells have a more quiescent metabolism phenotype when compared to HCT116-GSK3β-WT cells. Although knockout of GSK3β did not affect cell viability *in vitro*, growth assays indicated that HCT116-GSK3β-KO cells exhibited a reduced growth rate and growth potential when grown in adherent cell culture conditions (Fig 5C).

## Discussion

We used proteomic and cell-biological approaches to analyse the functional consequences of loss of GSK3β in colorectal cancer cells. Since the role of GSK3β in regulation of Wnt/β-catenin signalling in colorectal cancer is comparatively well understood, we have used colorectal cancer cells with constitutively stabilized and activated β-catenin to identify other GSK3β regulated processes. We used an isogenic homozygous GSK3β knockout model derived from the

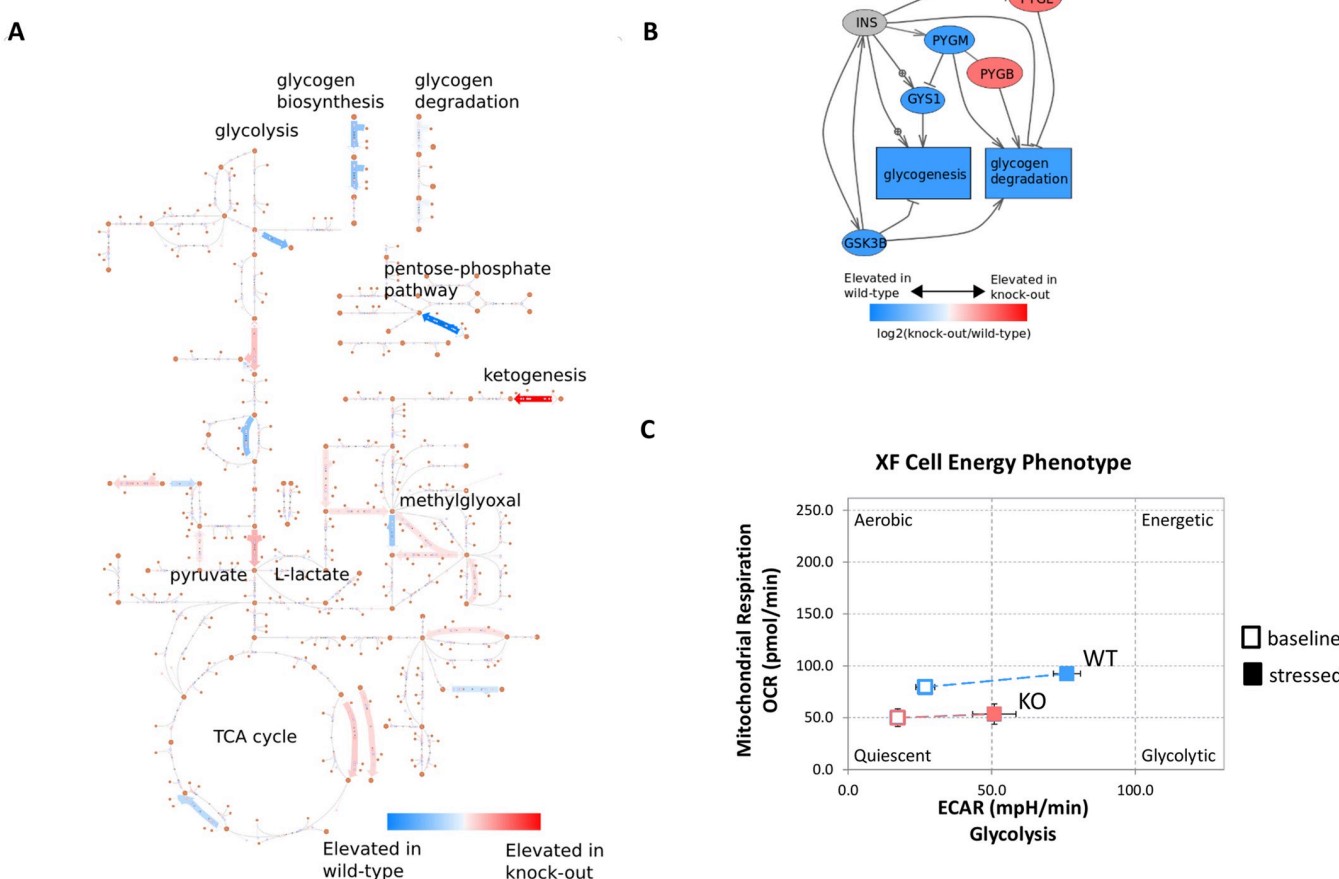

**Fig 5.** A. Map of metabolic pathway perturbations in HCT116-GSK3β-KO cells. All significantly differentially expressed proteins were mapped using the Escher tool for metabolic pathway mapping. B. Glycogen metabolism proteins significantly altered in HCT116-GSK3β-KO cells. Glycogen metabolism protein sub-network. All red or blue colored proteins were significantly (p<0.05) differentially expressed in wild-type (blue) or knockout cells (red). C. Metabolism phenotype assay of HCT116-GSK3β-WT and HCT116-GSK3β-KO cells. Cell metabolism assessed by Agilent Seahorse XFp cell energy phenotype test, oxygen consumption rate (OCR) and extracellular acidification rate (ECAR) were used as measures of mitochondrial respiration and glycolysis respectively, under basal and stressed conditions to assess metabolic potential of cells. N = 3.

model colorectal cancer cell line HCT116. We confirmed that both wild-type and GSK3β knockout HCT116 cells exhibit robust Wnt/β-catenin signalling by analysis of upstream and downstream components of the signalling pathway. Although the majority of Wnt related proteins we examined did not exhibit significant differential expression, we did observe several specific changes. Adenomatous Polyposis Coli (APC) is a key member of the β-catenin destruction complex, and in our studies was identified only in HCT116-GSK3β-WT cells (in 2 out of 3 samples), and not in HCT116-GSK3β-KO cells, indicating a likely reduction in the knockout cells. Notably, APC contains several phosphorylation consensus sequences for both GSK3β and CSK1, and GSK3β has been shown to mediate the interaction between APC and Axin in the destruction complex [20]. Although we did not observe a statistically significant alteration to β-catenin, α-catenin (CTNNA1) was significantly decreased in the HCT116-GSK3β-KO cells. GSK3β has been linked to phosphorylation of α-catenin in a double knockout phosphoproteomic study, where no phosphorylated α-catenin was detected in knockout cell models [21]. In addition, loss of α-catenin phosphorylation sites were linked to

reduced strength of cell-cell adherens junctions [22]. Taken together with the Western analysis and Wnt transcriptional activity assays, these results show that Wnt/β-catenin signalling was not significantly altered in HCT116-GSK3β-KO cells.

Our proteomic analysis identified substantial alterations to the expression of plakophilins and integrins involved in desmosome and hemidesmosome formation, protein complexes that mediate cell-cell and cell-substrate adhesion respectively. Hemidesmosomes bring about the adhesion of basal epithelial cells to the underlying basement membrane and consist of a multiprotein complex [23]. This multi-protein complex is anchored by integrin α6β4, and both α6 and β4 showed significant and substantial decreases in HCT116-GSK3β-KO cells. Integrins α6 and β4 were the only integrins identified in our study as significantly differentially expressed, consistent with the observation that in epithelial cells β4 only dimerizes with α6 subunit in an exclusive association [24]. We investigated the strength of cell-cell adhesion and found that HCT116-GSK3β-KO were significantly less adherent than HCT116-GSK3β-WT cell films consistent with the proteomic data indicating major hemidesmosome disruption.

We also observed differential levels of key components of the desmosomes, including plakophilins 2 and 3 (PKP2 and PKP3) and plakoglobin (JUP), all armadillo repeat proteins, in common with β-catenin. Consistent increased or decreased expression was observed for these 3 proteins in the mass-spectrometry analysis and Western blot analysis. PKP2 and PKP3 were both increased in HCT116-GSK3β-KO whereas plakoglobin was decreased. Desmoplakin (DSP), a core component of the desmosome interacts with plakoglobin and plakophilins. DSP was identified and quantified in all mass-spectrometry samples analysed, but showed no significant differential expression between KO and WT cells. GSK3β-mediated phosphorylation of the C-terminal of desmoplakin (DSP) is crucial for binding of its interaction partners, mainly cytoskeletal proteins and plakoglobins [25]. Loss of DSP phosphorylation by knockdown or knockout of GSK3β results in the formation of desmosomes lacking a connection to the cytoskeletal intermediate filaments, reducing the strength of desmosome cell-cell junctions [25]. GSK3 and PRMT1 have been shown to cooperatively catalyse post-translational modification of desmoplakin and inhibition of GSK3 resulted in delayed formation of adherent junctions [8], although with the caveat that in the latter study the authors do not differentiate between GSK3β and GSK3α. Intriguingly, upon knockdown of GSK3, DSP aggregates at interactin filaments in the cytoplasm which is mediated by GSK3 phosphorylation of DSP at S2849, without affecting the protein abundance of DSP [8]. This relocalization of the protein results in a disconnection of the desmosomal plaque from the intracellular cytoskeleton, reducing the strength of desmosomal plaques. Relocalization of DSP may occur in the HCT116-GSK3β-KO cells as a result GSK3β loss, explaining the observations we have made.

We also observed substantial alterations to metabolic pathway utilisation in HCT116-GSK3β-KO cells. Of the 10 most statistically significant proteins that are increased in HCT116-GSK3β-KO cells, 3 of them are associated with glucose metabolism (Pyruvate kinase is the 2nd most significantly altered protein in HCT116-GSK3β-KO cells). These findings are in line with the role that GSK3β plays in the regulation of glucose metabolism, whereby glycogen synthase is a substrate of GSK3β kinase activity [26]. A particular challenge in interpretation of the proteomic analysis is delineating the role that GSK3β plays in regulation of the observed metabolic pathways. This may be a direct role through phosphorylation as for glycogen synthase, or the perturbations observed could be indirectly associated with these direct alterations. Whilst our study does not identify direct substrates of GSK3β (which have been defined through multiple phosphoproteomic studies [21]), our quantitative proteomic analysis does identify the dysregulated protein networks, pathways and process and points to the multi-functional nature of GSK3β. Future studies that target the specific activities of GSK3β in cell-cell adhesion or regulation of metabolic activity may need more specific model systems

that target specific activities of GSK3β or the activity of GSK3β in a specific protein complex or sub-cellular localisation [27].

## Supporting information

**S1 Raw images. Underlying western blot images.**
(PDF)

**S1 File. Supplementary table of protein quantification of HCT116-GSK3β-WT and HCT116-GSK3β-KO cells by mass-spectrometry.**
(XLSX)

**S2 File. Supplementary table of enriched pathways from the mass-spectrometry analysis.**
(XLSX)

## Author Contributions

**Conceptualization:** Emily Bowler-Barnett.

**Funding acquisition:** Paul Skipp, Rob M. Ewing.

**Investigation:** Emily Bowler-Barnett, Francisco D. Martinez-Garcia, Matthew Sherwood, Ahood Aleidan, Steve John, Sara Weston.

**Methodology:** Emily Bowler-Barnett.

**Project administration:** Paul Skipp, Rob M. Ewing.

**Supervision:** Yihua Wang, Nullin Divecha, Paul Skipp, Rob M. Ewing.

**Validation:** Emily Bowler-Barnett.

**Writing – original draft:** Emily Bowler-Barnett, Paul Skipp, Rob M. Ewing.

**Writing – review & editing:** Emily Bowler-Barnett, Rob M. Ewing.

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
