## [Decision Letter · Decision Letter 0]

26 Mar 2021

PONE-D-20-40086

Proteomic characterization of GSK3β knockout shows altered cell adhesion and metabolic pathway utilisation in colorectal cancer cells

PLOS ONE

Dear Dr. Ewing,

Thank you for submitting your manuscript to PLOS ONE. After careful consideration, we feel that it has merit but does not fully meet PLOS ONE’s publication criteria as it currently stands. Therefore, we invite you to submit a revised version of the manuscript that addresses the points raised during the review process.

The two reviewers each have a substantial list of concerns and comments; please address each explicitly in a separate document and revise the manuscript accordingly.  that should be addressed explicitly.  

We look forward to receiving your revised manuscript.

Kind regards,

Michael Klymkowsky, Ph.D.

Academic Editor

PLOS ONE

Journal Requirements:

4. Thank you for submitting the above manuscript to PLOS ONE. During our internal evaluation of the manuscript, we found significant text overlap between your submission and the following previously published works, some of which you are an author.

https://www.tandfonline.com/doi/full/10.1080/15592294.2019.1656154?scroll=top&needAccess=true

https://eprints.soton.ac.uk/cgi/users/login?target=https%3A%2F%2Feprints.soton.ac.uk%2F434591%2F1%2FEmilyBowlerBarnett_PhD_thesis.pdf

Please revise the manuscript to rephrase the duplicated text, cite your sources, and provide details as to how the current manuscript advances on previous work. Please note that further consideration is dependent on the submission of a manuscript that addresses these concerns about the overlap in text with published work.

Reviewers' comments:

Reviewer's Responses to Questions

**Comments to the Author**

1. Is the manuscript technically sound, and do the data support the conclusions?

Reviewer #1: Partly

Reviewer #2: Partly

2. Has the statistical analysis been performed appropriately and rigorously? 

Reviewer #1: Yes

Reviewer #2: Yes

3. Have the authors made all data underlying the findings in their manuscript fully available?

Reviewer #1: Yes

Reviewer #2: Yes

4. Is the manuscript presented in an intelligible fashion and written in standard English?

Reviewer #1: Yes

Reviewer #2: Yes

5. Review Comments to the Author

Reviewer #1: This manuscript may be interesting since it describes potential involvment of GSK3B in metabolism, which so far has been demonstrated only for the alpha isoform not the beta one. In addition, it shows a role for GSK3B in regulating key components of the desmosomes. However, several points have to be addressed before acceptance. In addition, also basic formal adjustment have to be made, for example how to nominate proteins and their relative abbreviations. In addition, sometimes the discussion is a little superficial. For details see the attached document

Reviewer #2: In this study the authors address GSK3β-associated signalling pathways using human colorectal cancer cells HCT116 harbouring an activating mutation in the CTNNB1 gene. Although GSK3β has regulatory functions in Wnt signalling, GSK3β-deficient HCT116 cells did not exhibit any changes in the abundance of key Wnt signalling molecules, which is in line with previously published data. Using quantitative proteomic analysis, the authors identified changes in the abundance of proteins associated with intercellular communication, adhesion, and carbon metabolism. The authors observed significant changes in the amount of proteins forming desmosomes, hemidesmosomes and focal adhesions in GSK3β-deficient cells. The practical impact of these changes is presented in several functional assays in which GSK3β-deficient cells exhibited reduced resistance to mechanical agitation and increased migratory ability. Finally, the authors show that GSK3β-deficient cells displayed reduced mitochondrial respiration, glycolysis, and growth.

The results are interesting and novel; however, there are certain shortcomings in the text and figures (see below). After revision I would recommend the study for publication.

Major issues

1. The quality of figures is insufficient – the image resolution should be higher and the text better readable (especially Figure 1b – gene names and legend; Figure 2a – protein names; Figure 4a and 4b – photo description; Figure 5c – text inside the graph).

2. The authors should be more specific when describing the generation of GSK3β knock-out in HCT116 cells. First, it is not clear whether the HCT116-GSK3β-KO cell line originated from a mixed cell culture or whether it was derived from a single cell. Second, targeting of the GSK3β locus is described insufficiently. The authors should provide sequences of the GSK3β locus and ssDNA oligonucleotide along with the information about the TALENs used (target site sequence, catalogue number of plasmids). Finally, it is not apparent what are the "replicated samples". The authors do not explain whether the replicates represent cell lines targeted by different TALENs or different cell clones targeted by the same TALEN. This should be mentioned in the methods.

3. The principle of the TopFLASH assay and metabolic profiling (Agilent Seahorse XFp cell energy phenotype test) should be described, either in the (supplementary) methods or results. Alternatively, the authors could add references to papers describing these methods.

4. On page 12, some results of the mass-spectrometry analysis are mentioned, specifically β-catenin, APC and CK1α expression, but there is no reference to any figure. I understand that the data are presented in the Supplementary Table; however, this table is very extensive and difficult to search through. I suggest that the authors create an extra table containing only expression of those proteins that are discussed in the text.

5. In Figure 2, the protein expression in Western blot 2b is shown in three samples: "WT", "KO", and "HCT116" – it is not clear what is the third column, as both the "WT" and

"KO" are HCT116 cells. Next, Western blots in 2c and 2d display protein abundance in two "WT" and two "KO", while in 2e and 2f there is only one column for "WT" and "KO". The authors should clarify whether these figures display representative or all results. Finally, graphs in 2c and 2d lack y-axis description.

6. The authors evaluate expression of the Axin protein (page 12 and Figure 2e); however, they don’t specify whether it is Axin1 or Axin2. In addition, the anti-Axin antibody should be added into methods.

7. The authors claim that they identified differential abundance of proteins included in focal adhesion assembly (p=0.005) – page 14, lines 1-3; however, there is no reference to any figure or table supporting this claim. I suggest that the authors add these results (possibly within supplementary materials).

8. On page 14, a specific reduction of several proteins is described in HCT116-GSK3β-KO with reference to Figure 3A; however, the desmoglein protein, which is mentioned in the results, is not shown in the Figure. It should be added into the graph.

9. On page 16, the authors describe differential abundance of proteins involved in glycogen metabolism (PYGM, PYGL, PYGB, glycogen synthase 1) with a reference to Figure 5B; however, the figure contains only a scheme of the glycogen metabolism with color-coded protein abundance in WT or KO cells. I would be appropriate to show a graph or table with specific protein expression values as well.

10. In Figure 2, the Western blot analysis of WT cells is displayed in the first column and for KO cells in the following one, while in Figure 3, the first column in the graph shows protein abundance in KO cells and the second in WT cells, which is confusing for the reader. I suggest that the authors unify, in all figures, the order in which they show the results for WT and KO cells.

11. There is no reference to Figure 5D in the text. In addition, the y-axis description is not clear – the authors claim that they seeded cells at density 1 × 105 (page 17); however, the numbers on the y-axis named "Cell number" are in the range 0-800, which is confusing. This should be corrected.

Minor issues

12. The language is clear and easy to follow; however, there are several typos in the text (e. g. "6-cell" – page 7; "-20C" – page 20; "CSK1" – page 18) or missing words (e. g. Western analysis – page 18); therefore, I suggest that the authors have the text corrected by a copyeditor.

13. It would be appropriate to unify some terms and descriptions. For example, "β-Catenin" should be written with lowercase "c"; in "Wnt/ β-catenin signalling", no extra space should follow the slash; in Figure 1B description, the sign of inequality "greater than" should be expressed by the symbol ">"; "P value" on page 13 and "P-value" on page 15 should be written as "p-value"; "alpha-catenin" on page 18 should be written as "α-catenin".

14. The names of the figures should describe their content, not the method used. For example in Figure 4B, instead of "Cell migration assay", more appropriate name would be "GSK3β-deficient HCT116 cells display increased migration potential". I suggest that the authors rename all the figures accordingly.

15. Some abbreviations are not explained in figure description (e.g. Figure 2A and 5B) or are explained more than once in the text (desmoplakin on page 14 and 19). I ask the authors to correct this.

16. On page 6, a Tbc1d4 antibody is listed in the table, but there is no mention of the Tbc1d4 protein in the text.

17. I ask the authors to check whether the Figure numbers correspond with references in the text – e.g. on page 12, there is a reference to Figure 1G, which should be numbered 2G.

18. According to the text on page 14, Figure 3 is divided to Figure 3A (mass spectrometry analysis) and Figure 3B (Western blot analysis). This division is lacked in the Figure itself, as the letter "B" is missing in both, the Figure and the figure description. I ask the authors to unify this.

6. PLOS authors have the option to publish the peer review history of their article (what does this mean?). If published, this will include your full peer review and any attached files.

Reviewer #1: No

Reviewer #2: **Yes: **Monika Stastna

---

## [Author Response · Author response to Decision Letter 0]

20 Sep 2021

We thank the reviewers for their insightful comments and for helping us improve the manuscript. Alongside requested changes to the manuscript we have performed new experiments. These include new Western blots of new proteins showing the dys-regulation of proteins associated with cell-cell and cell-substrate adhesion. We also have redone the soft agar colony growth assay in line with the comments of the reviewers.

We address each point in turn below (all of our responses are in bold).

List of modified figures/tables:

Figure 1B (gene symbol labels removed)

Figure 2A (shading changed)

Figure 2E (Axin->Axin2)

Figure 3A (new data)

Figure 3C (new data)

Figure 3D (new data)

Figure 4C (new data)

Figure 5C (legend/text)

Figure 5D (removed)

Supplementary Table 1 (tab added)

Supplementary Table 2 (new)

Reviewer #1

This manuscript may be interesting since it describes potential involvement of GSK3B in metabolism, which so far has been demonstrated only for the alpha isoform not the beta one. In addition, it shows a role for GSK3B in regulating key components of the desmosomes. However, several points have to be addressed before acceptance. In addition, also basic formal adjustment have to be made, for example how to nominate proteins and their relative abbreviations. In addition, sometimes the discussion is a little superficial. For details see the attached document

Page 3, Abstract

- Consistent with the role GSK3β, we found that… -> Consistent with the role of GSK3β, we found that….

- …therefore use this knockout cell model ->… therefore used this knockout cell model

- …and we characterize the phenotype and altered proteomic profiles associated with this -> ….and we characterized the phenotype and altered proteomic profiles associated with this

- We also characterize the perturbation…-> We also characterized the perturbation…

- …and identify defects -> and identified defects

Materials and methods

- Several times the 6-well plates have been indicated as 6-cell plates

- 

Pag.12

- We tested the canonical Wnt-driven transcriptional activity of these cells through reporter (TopFLASH) analysis as shown in Figure 1G.

It’s Figure 2G

- Finally, we identified additional Wnt signaling components in the mass-spectrometry data. β-catenin showed no significant difference between WT and KO cells, whilst 

- All of the above comments have been addressed.

- 

- APC (Adenomatous Polyposis Coli), an important component of the destruction complex was only identified in WT cells, indicating that loss of GSK3β may result in loss or reduced expression of APC. Finally, the Casein Kinase 1 alpha subunit (CSNK1A1) which phosphorylates β-catenin at S45 showed reduced expression in knockout cells (p-value = 0.013).

The Authors show the western blot for all those proteins whose expression does not change in the absence of GSK3B but do not show the western blot for the only proteins whose expression is indeed changed: It’s a non-sense! So please add the blots for APC and CSNK1A1

We’ve steered the manuscript towards focusing more on the cell adhesion perturbations that we observed in the knock-out cells, in part because this is a more novel area than the role of GSK3B in Wnt signaling (which is well understood). We therefore include new Western blots for the cell-cell adhesion proteins of the desmosome and hemidesmosome, rather than expanding our analysis of the Wnt proteins. Whilst we agree with the author, that validation of proteins via Western blots that are identified by mass-spectrometry would be ideal for as many of the proteins as possible, in this instance, we feel that our efforts are best directed towards the cell-cell adhesion proteins (please see comments below for further explanation). To make this clearer for the reader, in Figure 2 we now highlight only those proteins in the Wnt pathway diagram (Figure 2A) that we have investigated by Western analysis. 

Pag.14

- The HCT116- GSK3β-KO cells showed a specific reduction in protein abundance of desmoplakin (DSP), plakoglobin (JUP) and desmoglein (DSG), and an increase in abundance of their essential linker proteins, plakophilins.

We also observed the complete lack of expression in HCT116-GSK3β-KO cells of two key integrin family members (Figure 3A). ITGA6 and ITGB4 

In the western blot of Fig.3 (that should be presented as a separate panel, 3B) not all of the proteins identified ad differentially expressed in KO cells by MS are validated and in the text is mentioned a protein, desmoglein, whose expression is not presented in the figure neither as MS nor as WB data. Please show the expression of all the proteins both by MS and WB. In addition, remember that there are rules as of proteins have to be indicated: full names followed by acronyms in brackets should be indicated in the text the first time a protein is mentioned and acronyms can be used in the figures but they have to explained in the legend. So please correct the text and the legend accordingly. In fact, for desmoplakin and plakoglobin the acronym is indicated in the text but for plakophilins it is not. Then the Authors report of the complete loss of expression of two key integrin family members but they don’t specify who they are….???? It’s up to the reader to guess they are ITGA6 and ITGB4 whose function is explained in the following sentence. By the way, these two proteins are indicated by acronyms without displaying the full name…. Finally, in figure 3 proteins are indicated by their full name for MS experiments and by acronyms in the WB panel!!... that are not explained in the figure legend… this is extremely confusing for the reader! 

We provide new Western blots of the desmoglein protein identified in the mass-spectrometry (DSG2) and of integrin A6 (ITGA6). We have also made it clearer in the text which of the proteins we have analysed by Western, including canonical gene symbols for clarity.

- Following mechanical agitation of detached monolayers of HCT116-

GSK3β-KO cells the cell monolayer fragmented into numerous pieces, in contrast to the HCT116-GSK3β-WT cells in which the monolayer remained intact

Is it possible a quantification of the phenomena for example, by counting the fragments in a given area?

The dispase cell-cell adhesion assay allows for the qualitative assessment of the disruption of cell-cell adhesion, and is typically paired (e.g. Roberts et al, PLoS One, Vol 8, 10, 2013) with scratch/wound assays as we have done to show impairment of cell-cell adhesion. Truly quantitative analysis of the strength of cell-cell adhesion would be achieved though more sensitive single-cell techniques such as atomic force microscopy. The number of fragments could be counted per plate well and compared between the cell-lines. However, we don’t believe that this pseudo-quantitative analysis would significantly add to the qualitative analysis that we have performed, nor change our findings that the desmosome function in HCT116-GSK3KO cells are grossly altered and this has changed the cell-matrix and cell-cell adhesion properties of the cells. 

Pag.15

- a soft agar colony growth assay was performed and colony number over 6

days recorded (Fig 4C). Results indicate that loss of GSK3β did not incur any colony-forming advantage to HCT116-GSK3β-KO cells 

First of all, usually the results of a soft agar assay are shown as images of the cristal violet-stained welsl accompanied by the graph showing the quantification. Only the graph is not acceptable.

Second, the soft agar assay is an endpoint assay ie, colony forming in two different experimental situation are counted at the end of the experiment only, and not every day. This because, by definition, a colony is considered when it is formed by at least 50 cells: given that HCT116 have a cell cycle of approximately 24hs it means that a colony is formed between day 6 and day 7. 

Therefore I request a “real”experiment of soft agar with representative images of the colonies after 7-10 days, accompanied by quantification graphically shown

We have completely redone this experiment as the reviewer has requested and now show the colony formation across the 7-10 day period with representative images.

Finally, what do you mean for: 

- although we observed an increased cell-count in cultures of HCT116-GSK3β-WT as compared to HCT116-GSK3β-KO (Figure 4C).

Reading this sentence I understand that the Authors are referring to a different experiment like a growth curve (which is actually performed in cultures) not to a soft agar assay. Please explain

We’ve now clarified this by explaining the colony formation results as shown in the new Figure.

Figure 5 

Legend of panel C 

There’s no legend of the graph: please indicate what the colors and empty/filled squares means 

We’ve now added the legend to Figure 5C.

Growth curve shown in panel D 

From day 6 to day 9 a plateau for WT cells and a decrease in survival of KO cells is evident, likely to be due to nutrients exhaustion. To assess the growth potential of a cell line cells should be plated at 10% and then counted every 24 hs in the log phase that in this experiment appears to be from day 1 to 6. Why the Authors decided to carry on for 3 more days? They exposed cells to an “artificial” stress in the final part of the experiment, so the apparent reduced “growth potential” (as they indicate it at page 17) of KO cells is in reality an a decline in the number of cells due to increased death of cells in conditions of nutrient exhaustion. WT cells seem instead to cope better in face of such a stress since the decrese in the number of cells is minimal. In order to get reliable and interpretable results the Authors should repeat the experiment in better controlled conditions such as stopping the growth count at 6 days. It is clear from the data they showed that WT cells are more resistant to limited nutrient availability so I would perform more experiments in that sense ie, repeating growth curves also maintaining cells in reduced amount of serum (eg in 1% serum) or assessing their viability in absence of serum by FACS analysis or by CellTiter-Glo® Luminescent Cell Viability Assay or even by a very simple Trypan blue staining.

The reviewer points out that the main difference between the growth profiles of the WT and KO cells is the period after 6 days where there is likely to be nutrient exhaustion. During the log phase, there is little difference between the cell-lines. We have therefore removed this figure from the results since we do not believe this result will add to the overall findings. 

Pag.19

- Loss of DSP phosphorylation by knockdown or knockout of GSK3β results in the formation of desmosomes lacking a connection to the cytoskeletal intermediate filaments, reducing the strength of desmosome cell-cell junctions (24). GSK3 and PRMT1 have been shown to cooperatively catalyse posttranslational modification of desmoplakin and inhibition of GSK3 resulted in delayed formation of adherent junctions (8). Intriguingly, upon knockdown of GSK3β, DSP aggregates at interactin filaments in the cytoplasm which is mediated by GSK3β phosphorylation of DSP at S2849, without affecting the protein abundance of DSP (8). This relocalization of the protein results in a disconnection of the desmosomal plaque from the intracellular cytoskeleton, reducing the strength of desmosomal plaques. Relocalization of DSP may occur in the HCT116-GSK3β-KO cells as a result GSK3β loss, explaining the observations we have made.

In this paragraph the Authors refer indiscriminately to GSK3 and GSK3β but this is not correct since the alpha and beta isoforms, although being largely redundant play also unique roles as demonstrated by gene knockout studies. In fact, GSK3A is unable to rescue the lethal phenotype of GSK3B null mice: the animals die during embryogenesis as a result of liver degeneration caused by widespread hepatocyte apoptosis, where excessive TNF-alpha-mediated cell death occurs, due to reduced

NFkB function (Hoeflich Ket al., 2000, Nature 406: 86–90). On the other hand, GSK3A null mice are viable and show metabolic defects – such as enhanced glucose and insulin sensitivity and reduced fat mass - which cannot be counteracted by the beta isofom (MacAulay K, et al.,2007, Cell Metab 6: 329–337). The reference used here to support their reasoning is vague about which isoform is capable to mediate the effects described since these Authors mostly use in their experiments LiCl which is a inhibitor of both isoforms. Therefore, it would be better to find an appropriate reference or modify the discussion. 

We recognize that in the Albrecht et al paper that we cited, the authors did not use antibodies or techniques that discriminated between GSK3B and GSK3A. Given the findings though, it’s a very relevant paper for our discussion. We’ve therefore kept the citation but modified the discussion to acknowledge the point the reviewer makes. 

Pag.19

- These findings are in line with the extensive role that GSK3β plays in the regulation of metabolic pathways, including the early finding that glycogen synthase is a substrate of GSK3β kinase activity (25).

Also in this case the statement is not well supported by the citation quoted. In fact, the reference is the original paper describing the glycogen synthase as a substrate of GSK3β. What about papers supporting the extensive role that GSK3β plays in the regulation of metabolic pathways? To my knowledge is more GSK3A than GSK3B that plays a role in the metabolis, given that GSK3A null mice are viable and show metabolic defects which cannot be counteracted by the beta isofom (see above). Therefore please quote the data from the literature supporting this statement or modify the discussion. Moreover, given that the data from Authors indicate a role for GSK3B in metabolic regulation can they exclude that the knockout of the beta isoform do not trigger a compensatory upregulation of GSK3A? 

We’ve now altered the discussion to indicate the specific role of GSK3B in glucose metabolism, and we recognize that there is plenty of evidence of GSK3 isoforms and their role in regulation of metabolic pathways, but many of these studies may not differentiate between isoforms, or as the reviewer indicates point to GSK3A rather than GSK3B.

Reviewer #2: 

In this study the authors address GSK3β-associated signalling pathways using human colorectal cancer cells HCT116 harbouring an activating mutation in the CTNNB1 gene. Although GSK3β has regulatory functions in Wnt signalling, GSK3β-deficient HCT116 cells did not exhibit any changes in the abundance of key Wnt signalling molecules, which is in line with previously published data. Using quantitative proteomic analysis, the authors identified changes in the abundance of proteins associated with intercellular communication, adhesion, and carbon metabolism. The authors observed significant changes in the amount of proteins forming desmosomes, hemidesmosomes and focal adhesions in GSK3β-deficient cells. The practical impact of these changes is presented in several functional assays in which GSK3β-deficient cells exhibited reduced resistance to mechanical agitation and increased migratory ability. Finally, the authors show that GSK3β-deficient cells displayed reduced mitochondrial respiration, glycolysis, and growth.

The results are interesting and novel; however, there are certain shortcomings in the text and figures (see below). After revision I would recommend the study for publication.

Major issues

1. The quality of figures is insufficient – the image resolution should be higher and the text better readable (especially Figure 1b – gene names and legend; Figure 2a – protein names; Figure 4a and 4b – photo description; Figure 5c – text inside the graph).

We’ve now addressed this. Gene symbol labels have been removed from Figure 1b (there are simply too many to display in a readable way). Label size increased where possible.

2. The authors should be more specific when describing the generation of GSK3β knock-out in HCT116 cells. First, it is not clear whether the HCT116-GSK3β-KO cell line originated from a mixed cell culture or whether it was derived from a single cell. Second, targeting of the GSK3β locus is described insufficiently. The authors should provide sequences of the GSK3β locus and ssDNA oligonucleotide along with the information about the TALENs used (target site sequence, catalogue number of plasmids). Finally, it is not apparent what are the "replicated samples". The authors do not explain whether the replicates represent cell lines targeted by different TALENs or different cell clones targeted by the same TALEN. This should be mentioned in the methods.

We’ve now made it clearer that the HCT116-GSK3β-KO and paired isogenic control cell line HCT116-GSK3β-WT are commercially acquired cell-lines (i.e. the cell-lines were not engineered by our laboratory) – they were purchased by us.

3. The principle of the TopFLASH assay and metabolic profiling (Agilent Seahorse XFp cell energy phenotype test) should be described, either in the (supplementary) methods or results. Alternatively, the authors could add references to papers describing these methods.

We’ve now added an appropriate reference for a methods article for Topflash and point the reader to the Agilient website for a description of how the Agilent seahorse works.

4. On page 12, some results of the mass-spectrometry analysis are mentioned, specifically β-catenin, APC and CK1α expression, but there is no reference to any figure. I understand that the data are presented in the Supplementary Table; however, this table is very extensive and difficult to search through. I suggest that the authors create an extra table containing only expression of those proteins that are discussed in the text.

Please see the response made to Reviewer 1’s comments concerning APC and CSNK1A. We’ve added more to the data concerning the cell-cell adhesion proteins and do not emphasize those proteins for which we have not validated via western blot. We think that this provides a better focus for the manuscript. Also, we have not added another table of MS results to the paper – but rather added an additional tab to Supplementary Table 1 where we list the key proteins referenced in the paper with all of the mas-spectrometry intensity data.

5. In Figure 2, the protein expression in Western blot 2b is shown in three samples: "WT", "KO", and "HCT116" – it is not clear what is the third column, as both the "WT" and

"KO" are HCT116 cells. Next, Western blots in 2c and 2d display protein abundance in two "WT" and two "KO", while in 2e and 2f there is only one column for "WT" and "KO". The authors should clarify whether these figures display representative or all results. Finally, graphs in 2c and 2d lack y-axis description.

We’ve now clarified these points in the figure image.

6. The authors evaluate expression of the Axin protein (page 12 and Figure 2e); however, they don’t specify whether it is Axin1 or Axin2. In addition, the anti-Axin antibody should be added into methods.

It is Axin 2 as now indicated in the figure, and with the antibody listed in the table.

7. The authors claim that they identified differential abundance of proteins included in focal adhesion assembly (p=0.005) – page 14, lines 1-3; however, there is no reference to any figure or table supporting this claim. I suggest that the authors add these results (possibly within supplementary materials).

We’ve added an additional Supplementary table (Supplementary Table 2) with the pathay analysis and the proteins within each group.

8. On page 14, a specific reduction of several proteins is described in HCT116-GSK3β-KO with reference to Figure 3A; however, the desmoglein protein, which is mentioned in the results, is not shown in the Figure. It should be added into the graph.

A new Western blot showing the desmoglein protein identified in the mass-spectrometry (DSG2) has been added (Figure 3C).

9. On page 16, the authors describe differential abundance of proteins involved in glycogen metabolism (PYGM, PYGL, PYGB, glycogen synthase 1) with a reference to Figure 5B; however, the figure contains only a scheme of the glycogen metabolism with color-coded protein abundance in WT or KO cells. I would be appropriate to show a graph or table with specific protein expression values as well.

We’ve now added these as an addendum to Supplementary Table 1, so that readers can see the protein expression values for the selected proteins discussed in the paper.

10. In Figure 2, the Western blot analysis of WT cells is displayed in the first column and for KO cells in the following one, while in Figure 3, the first column in the graph shows protein abundance in KO cells and the second in WT cells, which is confusing for the reader. I suggest that the authors unify, in all figures, the order in which they show the results for WT and KO cells.

All Westerns are now in the order WT then KO.

11. There is no reference to Figure 5D in the text. In addition, the y-axis description is not clear – the authors claim that they seeded cells at density 1 × 105 (page 17); however, the numbers on the y-axis named "Cell number" are in the range 0-800, which is confusing. This should be corrected.

Please see comments to Reviewer 1 concerning Figure 5D.

Minor issues

12. The language is clear and easy to follow; however, there are several typos in the text (e. g. "6-cell" – page 7; "-20C" – page 20; "CSK1" – page 18) or missing words (e. g. Western analysis – page 18); therefore, I suggest that the authors have the text corrected by a copyeditor.

These issues have now been addressed.

13. It would be appropriate to unify some terms and descriptions. For example, "β-Catenin" should be written with lowercase "c"; in "Wnt/ β-catenin signalling", no extra space should follow the slash; in Figure 1B description, the sign of inequality "greater than" should be expressed by the symbol ">"; "P value" on page 13 and "P-value" on page 15 should be written as "p-value"; "alpha-catenin" on page 18 should be written as "α-catenin".

14. The names of the figures should describe their content, not the method used. For example in Figure 4B, instead of "Cell migration assay", more appropriate name would be "GSK3β-deficient HCT116 cells display increased migration potential". I suggest that the authors rename all the figures accordingly.

15. Some abbreviations are not explained in figure description (e.g. Figure 2A and 5B) or are explained more than once in the text (desmoplakin on page 14 and 19). I ask the authors to correct this.

We’ve now taken care of the typos and inconsistencies, but have left the figure headings largely unchanged, because we think these shorter titles are sufficiently clear.

16. On page 6, a Tbc1d4 antibody is listed in the table, but there is no mention of the Tbc1d4 protein in the text.

17. I ask the authors to check whether the Figure numbers correspond with references in the text – e.g. on page 12, there is a reference to Figure 1G, which should be numbered 2G.

18. According to the text on page 14, Figure 3 is divided to Figure 3A (mass spectrometry analysis) and Figure 3B (Western blot analysis). This division is lacked in the Figure itself, as the letter "B" is missing in both, the Figure and the figure description. I ask the authors to unify this.

All these have now been addressed.

---

## [Decision Letter · Decision Letter 1]

4 Oct 2021

Proteomic characterization of GSK3β knockout shows altered cell adhesion and metabolic pathway utilisation in colorectal cancer cells

PONE-D-20-40086R1

Dear Dr. Ewing,

We’re pleased to inform you that your manuscript has been judged scientifically suitable for publication and will be formally accepted for publication once it meets all outstanding technical requirements.

Kind regards,

Michael Klymkowsky, Ph.D.

Academic Editor

PLOS ONE

Additional Editor Comments (optional):

Reviewers' comments:

Reviewer's Responses to Questions

**Comments to the Author**

1. If the authors have adequately addressed your comments raised in a previous round of review and you feel that this manuscript is now acceptable for publication, you may indicate that here to bypass the “Comments to the Author” section, enter your conflict of interest statement in the “Confidential to Editor” section, and submit your "Accept" recommendation.

Reviewer #1: All comments have been addressed

Reviewer #2: All comments have been addressed

2. Is the manuscript technically sound, and do the data support the conclusions?

Reviewer #1: Yes

Reviewer #2: Yes

3. Has the statistical analysis been performed appropriately and rigorously? 

Reviewer #1: Yes

Reviewer #2: Yes

4. Have the authors made all data underlying the findings in their manuscript fully available?

Reviewer #1: Yes

Reviewer #2: Yes

5. Is the manuscript presented in an intelligible fashion and written in standard English?

Reviewer #1: Yes

Reviewer #2: Yes

6. Review Comments to the Author

Reviewer #1: I feeel satisfied with the improvements made by the Authors in response to my comments, therefore I recommend the manuscript for publication

Reviewer #2: The authors edited the manuscript based on the comments from both reviewers and supplemented it with missing information. In its current form, I recommend the manuscript for publication.

One minor note: the term "glycogen-specific kinase" is used in the text, however, GSK3β acronym stands for "glycogen synthase kinase" - the authors should correct this.

7. PLOS authors have the option to publish the peer review history of their article (what does this mean?). If published, this will include your full peer review and any attached files.

Reviewer #1: No

Reviewer #2: No

---

## [Editor Report · Acceptance letter]

20 Oct 2021

PONE-D-20-40086R1 

Proteomic characterization of GSK3β knockout shows altered cell adhesion and metabolic pathway utilisation in colorectal cancer cells. 

Dear Dr. Ewing:

I'm pleased to inform you that your manuscript has been deemed suitable for publication in PLOS ONE. Congratulations! Your manuscript is now with our production department. 

Kind regards, 

on behalf of

Dr. Michael Klymkowsky 

Academic Editor

PLOS ONE